# *Vibrio parahaemolyticus* Isolates from Asian Green Mussel: Molecular Characteristics, Virulence and Their Inhibition by Chitooligosaccharide-Tea Polyphenol Conjugates

**DOI:** 10.3390/foods11244048

**Published:** 2022-12-14

**Authors:** Suriya Palamae, Ajay Mittal, Mingkwan Yingkajorn, Jirakrit Saetang, Jirayu Buatong, Anuj Tyagi, Prabjeet Singh, Soottawat Benjakul

**Affiliations:** 1International Center of Excellence in Seafood Science and Innovation, Faculty of Agro-Industry, Prince of Songkla University, Hat Yai, Songkhla 90110, Thailand; 2Department of Pathology, Faculty of Medicine, Prince of Songkla University, Hat Yai, Songkhla 90110, Thailand; 3College of Fisheries, Guru Angad Dev Veterinary and Animal Sciences University, Ludhiana 141004, India

**Keywords:** antibacterial, COS polyphenol conjugate, *Perna viridis*, *Vibrio parahaemolyticus*, virulence factor

## Abstract

Fifty isolates of *Vibrio parahaemolyticus* were tested for pathogenicity, biofilm formation, motility, and antibiotic resistance. Antimicrobial activity of chitooligosaccharide (COS)-tea polyphenol conjugates against all isolates was also studied. Forty-three isolates were randomly selected from 520 isolates from Asian green mussel (*Perna viridis*) grown on CHROMagar^TM^ Vibrio agar plate. Six isolates were acquired from stool specimens of diarrhea patients. One laboratory strain was *V. parahaemolyticus* PSU.SCB.16S.14. Among all isolates tested, 12% of *V. parahaemolyticus* carried the *tdh*^+^*trh*^−^ gene and were positive toward Kanagawa phenomenon test. All of *V. parahaemolyticus* isolates could produce biofilm and showed relatively strong motile ability. When COS-catechin conjugate (COS-CAT) and COS-epigallocatechin-3-gallate conjugate (COS-EGCG) were examined for their inhibitory effect against *V. parahaemolyticus*, the former showed the higher bactericidal activity with the MBC value of 1.024 mg/mL against both pathogenic and non-pathogenic strains. Most of the representative Asian green mussel *V. parahaemolyticus* isolates exhibited high sensitivity to all antibiotics, whereas one isolate showed the intermediate resistance to cefuroxime. However, the representative clinical isolates were highly resistant to nine types of antibiotics and had multiple antibiotic resistance (MAR) index of 0.64. Thus, COS-CAT could be used as potential antimicrobial agent for controlling *V. parahaemolyticus*-causing disease in Asian green mussel.

## 1. Introduction

*Vibrio parahaemolyticus* has become a serious foodborne pathogen and raised public health concern in Thailand, China, Japan, and other Asian countries [1]. *V. parahaemolyticus* present in aquatic products contributes to significant economic losses across the entire supply chain [2]. It is a member of the genus *Vibrio* from family *Vibrionaceae*, a Gram-negative, marine halophilic bacterium that naturally inhabits in global coastal waters, sediment and various types of marine animals [3] such as fish, shrimp, crab, clam, oyster [4,5,6,7], and mussel [8]. *V. parahaemolyticus* can also be transmitted to humans when consuming contaminated raw or poorly cooked seafood products [9,10]. This bacterium contributes to an acute gastroenteritis, which includes nausea, diarrhea, vomiting, fever, and chills. It can also cause severe symptoms in children, the elderly, and immunocompromised patients [11,12]. The pathogenic *V. parahaemolyticus* often infects and causes disease by using several virulence factors. Adhesions, thermostable direct hemolysin (TDH), and TDH-related hemolysin (TRH) are the most important virulence factors in this bacterium [13]. The hemolysis-associated genes, *tdh*, were found in the most multidrug-resistant (MDR) isolate as well as those having certain virulence characteristics and biofilm capacity. It is able to attach to several surfaces and subsequently proliferate, in which a multicellular consortium with a three-dimensional structure can be formed. Such a biofilm can protect the cells toward environmental stress [14]. In addition, the augmented resistance to antimicrobials is achieved. Moreover, biofilms can release viable *V. parahaemolyticus* cells into the environment as well as foods [14]. The aforementioned factors augment the pathogenicity of *V. parahaemolyticus*, causing the severe public health problem globally.

Recently, seafood has gained popularity internationally due to its health benefits, resulting in the augmented production and domestic consumption in Southeast Asia. Asian green mussel, *Perna viridis*, is abundant along the coasts and estuaries of Asian-Pacific regions. It is also a widespread species found from the Persian Gulf to the Southwest Pacific and from Southern Japan to Papua New Guinea [15]. It is an important commercial aquaculture bivalve in Southeast Asian countries [16]. The prevalence and outbreaks of *V. parahaemolyticus* infection in mussels were documented [17,18].

Antibiotic therapy is the treatment commonly used for bacterial infections; however antibiotic-resistant bacteria have been a major concern worldwide [19]. Natural alternative approaches to control foodborne pathogens have been researched. In recent years, many antimicrobial compounds from seafood processing leftover have been reported to prevent the proliferation of several pathogenic bacteria [20,21]. Those included low-molecular-weight (MW) chitooligosaccharide (COS), a deacetylated form of chitosan. It shows non-toxicity, biodegradable, and biocompatible properties. COS has been modified through several processes, particularly via polyphenol grafting [20]. Several polyphenols have been utilized for the modification of COS to augment their bioactivities involving antioxidant, antimicrobial, anticancer, antihypertension, etc., [22,23,24]. Singh et al. [20] prepared COS-epigallocatechin-3-gallate conjugates (COS-EGCG) encompassing high antioxidant and antimicrobial activities. Recently, Mittal et al. [21] documented that COS-catechin conjugate (COS-CAT) showed superior antioxidant and antimicrobial activities to COS and other COS-polyphenol conjugates. COS-CAT possessed the greatest antimicrobial activity toward both Gram-negative and Gram-positive bacteria. In general, antimicrobial activity of COS and polyphenols was linked to bacterial cell wall disintegration via electrostatic interaction with their -OH and amino groups. Furthermore, alterations in microbial mRNA, DNA, and protein synthesis as induced by the diffused COS and polyphenols can cause cell death [20,21]. COS polyphenol conjugates have been reported to be effective against a variety of microorganisms. Nevertheless, there are no reports on its antimicrobial activity against *V. parahaemolyticus* isolated from Asian green mussel and clinical sample. In addition, drug sensitivity, virulence, and molecular characteristics of *V. parahaemolyticus* isolated from Asian green mussel collected from the south of Thailand have not been investigated.

## 2. Materials and Methods

### 2.1. Sample Collection and Bacterial Isolation

Asian Green mussels (*Perna viridis*) (Figure 1) were collected randomly from the different Asian green mussel farms, natural habitat and fresh markets located in the southern provinces of Thailand (Suratthani, Trang and Songkhla provinces). Provinces, types of places, and number of collected samples are given in Table 1. The samples were brought to the laboratory in polyethylene bags containing ice within 1 h. The samples were washed with sterilized distilled water for surface sterilization and shucked by aseptic technique. Briefly, 25 g of Asian green mussel samples were transferred into 225 mL of alkaline peptone water (APW) (polypeptone, 10 g/L; NaCl, 20 g/L; pH 8.6) and mixed with the aid of stomacher (Stomacher 400 Seaward medicals, Worthing, UK) at 230 rpm for 1 min. The mixture was incubated at 41.5 °C for 6–8 h. The APW-enriched culture was diluted from 10^4^ to 10^7^–fold with the APW. Subsequently, 100 μL of the diluted samples were spread on thiosulphate citrate bile salts sucrose agar plate (TCBS Agar; Oxoid, Thermo Fischer Scientific, Waltham, MA, USA), and CHROMagar^TM^ Vibrio agar plate (CHROMagar^TM^, Paris, France) was adopted for selection of *V. parahaemolyticus* isolates. These two agar plates were used for confirmation. The result from CHROMagar^TM^ Vibrio plates was mainly used for further experiments. Forty-three isolates with different colony colors from the total 520 of isolates on CHROMagar^TM^ Vibrio agar plate were randomly selected and then characterized and specified by the MALDI-Biotyper^®^ system (microflex LT; Bruker Daltonik GmbH, Bremen, Germany) [25]. Halophilism was also performed using NaCl-tryptone broth (T_1_N_0_, T_1_N_3_, T_1_N_6_, T_1_N_8_, and T_1_N_10_).

Fifty isolates used in this study were molecularly identified and confirmed for *V. parahaemolyticus*. Forty-three isolates were retrieved from the Asian green mussel, while the remaining six isolates were isolated from the stool specimens of diarrhea patients from Songklanagarind Hospital, Faculty of Medicine and one laboratory strain of *V. parahaemolyticus* PSU.SCB.16S.14 was gifted by the Food Safety Laboratory, Prince of Songkla University, Hat Yai, Thailand.

### 2.2. Polymerase Chain Reaction (PCR) Assay

Twenty microliters of glycerol stock of *V. parahaemolyticus* isolates (*n* = 50) was inoculated into 5 mL of Luria-Bertani (LB) broth (Merck, Burlington, MA, USA) containing 3% NaCl (*w*/*v*) and incubated at 37 °C for 16–18 h, followed by centrifugation (8000 × *g* for 5 min). The genomic DNA was isolated using a PureLink^TM^ Genomic DNA Mini Kit (Invitrogen, Thermo Fisher Scientific, Waltham, MA, USA). DNA concentration was measured with the aid of a NanoDrop spectrophotometry (Thermo Fisher Scientific, Waltham, MA, USA). PCR primers were synthesized via Integrated DNA Technologies (Singapore city, Singapore) as shown in Table 2. PCR reaction mixture comprised 5 μL of 4 × *Taq* PCR Mastermix (QIAGEN, Germantown, MD, USA), 2 μL of genomic DNA (50 ng/μL), 0.5 μL of primer pair solution (10 μM each), and 12 μL of Rnase free water. PCR was amplified under the selected conditions: pre-denaturation at 95 °C for 2 min, 30 cycles for denaturation at 95 °C for 5 s, annealing at 58 °C for 15 s, and extension at 72 °C for 10 s, and ending extension at 72 °C for 5 min [26]. PCR products were finally determined using 2% agarose gel electrophoresis.

### 2.3. Preparation of COS-Tea Polyphenol Conjugates Using Free Radical Grafting Method

COS-CAT and COS-EGCG conjugates were prepared using free radical grafting method as tailored by Mittal et al. [21]. First, pH of COS solution (1%, *w*/*v*) was adjusted to 5.0 using acetic acid (1 M). Simultaneously, 1 M H_2_O_2_ (4 mL) containing 0.10 g ascorbic acid were incubated (40 °C, 10 min) to generate hydroxyl radicals. Both solutions were then mixed and the mixture was incubated at room temperature for 1 h with continuous stirring. CAT and EGCG (10%, *w*/*w* of COS) were then added into the mixture and incubated for 24 h in dark, at room temperature. With dialysis against distilled water, the unbound CAT and EGCG were removed. COS-CAT and COS-EGCG conjugate powders were obtained after lyophilization of dialysates.

### 2.4. Determination of Minimum Inhibitory Concentration (MIC) and Minimum Bactericidal Concentration (MBC)

MIC and MBC of COS-CAT and COS-EGCG toward *V. parahaemolyticus* isolates were measured following the guidelines of Clinical and Laboratory Standards Institute (CLSI). Overnight culture (18–24 h) of *V. parahaemolyticus* isolate was adjusted to a final concentration of 10^8^ CFU/mL (corresponding to approximately 0.5 McFarland standard). The standardized suspension was then diluted by 200-fold with Mueller Hilton Broth (MHB) (Difco^TM^, Baltimore, MD, USA) supplemented with 3% NaCl (*w*/*v*), namely “diluent” to obtain the working concentration of 10^6^ CFU/mL. The COS-CAT and COS-EGCG powders were dissolved and diluted with deionized water [28]. Stock solution was subjected to two-fold dilution to attain the highest concentrations of 2.048 mg/mL and the lowest concentration of 0.004 mg/mL. One-hundred microliters of bacterial suspension and 100 μL of COS-CAT/COS-EGCG working solutions were mixed in each well, and incubated for 24 h at 37 °C. Thereafter, 20 μL of resazurin (0.09%) solutions was added for each well, and further incubated for 3 h at 37 °C. Subsequently, the wells with no color change were scored as “above the MIC value”. MBC was determined by plating directly the content of wells with concentration higher than the MIC value. Culture solution (10 μL) was pipetted from each well with no bacterial growth and dropped uniformly on a sterile MHA medium [29], followed by incubation (37 °C for 24 h). MBC was defined as the minimum concentration of COS-tea polyphenol conjugated solutions without colony formation. Positive control consisted of bacterial suspension and diluent, while negative control comprised MHB and diluent.

### 2.5. Biofilm Crystal Violet (CV) Staining

CV staining method was adopted [30]. Overnight cultures were diluted 50-fold using 200 μL of Oxoid TSB broth (Oxoid Ltd., Hampshire, England) containing 3% NaCl (*w*/*v*) (TSB-N) in 96-well plates (Corning Inc., Corning, NY, USA). Culture was allowed to proliferate at 37 °C for 48 h. The cultures were removed and the well with the adherent biofilm was gently washed with 200 μL of sterile phosphate buffered saline (PBS) for three times. Then, 200 μL of 0.1% crystal violet was used to stain the surface-attached cells for 15 min. After solution removal, the well was thoroughly washed with sterile H_2_O for three times. Bound dye in each well was solubilized using 200 μL of ethanol (Analytical grade ≥ 99.9% in pure, RCI Labscan™, Bangkok, Thailand). Absorbance at 570 nm (A_570_) was measured.

### 2.6. Analysis of Swimming and Swarming Motility

Swimming and swarming abilities of *V. parahaemolyticus* isolates were examined on semi-solid swimming plated (TSB-N in the presence of 0.2% agar) and solid swarming plates (TSB-N containing 0.5% agar), respectively [30,31]. The overnight cultures were diluted 50-fold using 5 mL of TSB-N broth and cultured at 37 °C with continuous shaking (200 rpm) until A_570_ reached 1.2–1.4. Those cultures were used for testing.

### 2.7. Kanagawa Phenomenon (KP) Test

KP test was performed as detailed by Zhang et al. [32]. First, 2 μL of the third-round cell cultures were inoculated onto Wagatsuma agar medium consisting of 5% rabbit red blood cells (RBCs). The radius from the inoculation place to the edge of β-hemolysin zone was detected after static incubation (37 °C for 24 h).

### 2.8. Antibiotic Susceptibility Testing

Antimicrobial susceptibility testing of eight *V. parahaemolyticus* isolated from clinical sample, Asian green mussel samples from different origins and laboratory strain were performed using the Sensititre^TM^ microbroth dilution system (Trek Diagnostic Systems, Cleveland, OH, USA) [33]. Cultures were grown overnight on TSA supplemented with 2.5% NaCl (*w*/*v*) plates at 37 °C. The cultures were transferred to sterile demineralized 2.5% saline solution to obtain the turbidity, equivalent to that of 0.5 McFarland standard. One-hundred milliliters of each suspension were transferred into sterile cation-adjusted MHB, and broth solution (50 mL) was dispersed onto CML1FMAR custom MIC plates (Trek Diagnostic Systems Inc., Cleveland, OH, USA) containing 21 different antibiotics with varying ranges of concentrations (μg/mL): amikacin (8–32), ampicillin (8–16), ampicillin/sulbactam (4/2–16/8), amikacin/clavulanic acid (4/2–16/8), cefepime (1–32), cefotaxime (1–32), cefoxitin (4–16), ceftazidime (1–32), ceftriaxone (0.5–32), cepfuroxime (8–16), ciprofloxacin (0.06–2), colistin (1–8), doripenem (0.5–16), ertapenem (0.5–4), gentamicin (2–8), imipenem (0.5–16), levofloxacim (0.06–8), meropenem (0.05–16), netilmicin (8–16), piperacillin-tazobactam (8/4–64/4), and sulfamethoxazole (1/19–4/76). MIC was the lowest concentration of the tested antibiotic, which totally inhibited bacterial growth [34]. Resistance breakpoints were also used [34]. Multiple antibiotic resistance (MAR) index was calculated as tailored by Krumperman [35], in which the following equation was used:MAR index = a/b
where “a” is the number of antibiotics, to which the particular isolate was resistant and “b” is the total number of antibiotics tested.

### 2.9. Statistical Analyses

Completely randomized design (CRD) was used for the entire study. The experiments and analyses were conducted in triplicate. Data were subjected to one-way analysis of variance (ANOVA) and a least significant difference test was used. *p* < 0.05 was considered a significant difference.

## 3. Results and Discussion

### 3.1. Characteristics of V. parahaemolyticus Isolates

All fifty collected isolates from Asian green mussel samples, clinical and laboratory strains, were identified as *V. parahaemolyticus* based on their morphological and biochemical characteristics. Double-plating method was used to identify species involving TCBS and CHROMagar™ Vibrio agars, the selective media providing the direct colony-color-based identification of *V. parahaemolyticus* by specific color development of the particular colonies. Out of 26 Asian green mussel collected samples, *V. parahaemolyticus* was detected in all the samples on TCBS agar (Figure 2A) and CHROMagar^TM^ Vibrio agar (Figure 2B). Colonies of fifty *V. parahaemolyticus* isolates appeared. All *V. parahaemolyticus* colonies were spherical, transparent, and bluish-green or green color on TCBS plates. On CHROMagar^TM^ plates, the colonies of fifty isolates were round, smooth, flat, mauve or purple red or purplish cream colony in color (positive colony = mauve color). On CHROMagar^TM^ Vibrio agar, the colony colors of 50 *V. parahaemolyticus* were varied. No.1 was a laboratory strain (PSU.SCB.16S.14); No. 2–44 were *V. parahaemolyticus* isolated from Asian green mussel; and No. 45–50 were *V. parahaemolyticus* isolated from clinical samples. Lee et al. [36] found that 4 (10.5%) of the 38 *V. parahaemolyticus* strains had white colonies on ChromoVP agar. Su et al. [37] documented that 5% of *V. parahaemolyticus* strains appeared as white colonies on Bio-Chrome Vibrio medium. High variability and differential colony colors were observed when culture-based techniques were used for seafood, clinical, and environmental samples. Hence, molecular confirmation must be conducted to ensure the accurate detection of *V. parahaemolyticus*. All the isolates were also confirmed by MALDI Biotyper^®^ analysis and thermolabile hemolysin encoded by the *tlh* gene as species marker. As shown in Figure 3A, 100% of the 49 isolates including a positive laboratory strain (PSU.SCB.16S.14) were *tlh*-positive. The salt tolerance test also showed that all the recovered strains required sodium ions for their growth in media supplemented with 1% NaCl up to 8%. The result was in agreement with that reported by Beleneva et al. [38]. However, the isolates should be collected from other provinces or different geographic locations to acquire more data, in which a variety and abundance of strains can be gained.

### 3.2. Virulence, Molecular and Biochemical Characteristics of V. parahaemolyticus Isolates

#### 3.2.1. Virulence Genes

All fifty isolates identified as *V. parahaemolyticus* by biochemical, MALDI-Biotyper^®^ system tests and confirmed by PCR were detected for the *tdh* and *trh* genes. DNA fragments of 269 and 500 bp in size were produced from the amplification of *V. parahaemolyticus* pathogenic *tdh* and *trh* genes, respectively (Table 2). Six out of fifty (12%) samples were positive for the *tdh* gene (*tdh*^+^*trh*^−^) (Figure 3B). However, the isolates of *V. parahaemolyticus* having both *tdh*^+^*trh*^+^ and *tdh*^−^*trh*^+^ were not detected in this study. Most *V. parahaemolyticus* clinical isolates had positive result for KP test (Figure 4), thus confirming the presence of hemolysin *tdh* and/or *trh* genes [39]. Most *V. parahaemolyticus* isolated from food and environment do not carry *tdh* and/or *trh* genes [39]. *V. parahaemolyticus* strains having *tdh* gene and strains possessing both *tdh* and *trh* genes were found at very low level in mussel [40,41]. *Vibrio* species was isolated from bivalves and the culture environments along the Gyeongnam coast in Korea [42]. One hundred and ninety isolates of *V. parahaemolyticus* from oyster, mussel, and ark shell were negative for the *tdh* virulence genes, while 18 (9.5%) isolates were positive for *trh* virulence genes. All strains were positive for the *trh* gene when isolated from only oyster samples [42]. No *trh*^+^
*V. parahaemolyticus* strains was detected in warm climate, including Thailand. Rodriguez-Castro et al. [43] found that *trh*^+^ strains were dominant in the cold water, whereas *tdh*+ *V. parahaemolyticus* disseminated in warm water. Only clinical *V. parahaemolyticus* strains carried the *trh*^+^ genes. Bhoopong et al. [44] documented that only 0.5% (3/629) of the clinical *V. parahaemolyticus* isolates from the 63 patients in Thailand carried the *trh* gene alone, whereas 87.4% (550/629) and 7% (44/629) of the isolates possessed the *tdh* gene and both genes, respectively. Chen et al. [45] found that 93% and 1% of the 501 clinical *V. parahaemolyticus* isolates from southeastern China carried *tdh* and *trh* genes, respectively. Nevertheless, distributions of *tdh*^+^ and/or *trh*^+^ strains may vary, depending on detection method, sample sources and geographical origin [46].

#### 3.2.2. Hemolytic Activity

KP test was used to determine hemolytic activity of the isolates on the Wagatsuma agar containing 5% RBCs as depicted in Figure 4. Based on KP, the pathogenic isolates of *V. parahaemolyticus* could be differentiated from non-pathogenic counterpart. When bacterium lyses human erythrocytes, a pore-forming toxin known as the thermostable direct hemolysin (TDH) was produced. As shown in Figure 4, no Asian green mussel isolates showed β-hemolysis, whereas all of clinical isolates exhibited β-hemolysis; the latter were isolated from the stool of patients. All the *tdh*^+^*trh*^−^ isolates displayed a positive reaction as evidenced by a β-hemolysis zone surrounding the growth spot, whereas all the *tdh*^−^*trh*^−^ isolates showed the negative reaction. Although four isolates namely M6, M13, M48 and M58 from Asian green mussel exhibited weak hemolysis (Figure 4), none of them exhibited strong β-hemolysis. This weak hemolysis might be related with other virulence factors, apart from TDH orTRH. Strains, which produce few extracellular enzymes, could have the weak hemolysis [47]. Although no isolates showed β-hemolysis activity, the potential risk involved in consuming Asian green mussel must be taken into consideration because of its short generation time.

#### 3.2.3. Motility Ability

*V. parahaemolyticus* has dual flagellar systems, i.e., a single polar flagellum for swimming in liquid and peritrichous lateral flagella for swarming on surfaces [48]. In the present study, swimming and swarming of clinical and Asian green mussel isolates were compared. Mobility abilities of 50 isolates could be classified into three levels: weak, medium, and strong, which respectively indicated that their mobilities were much lower, similar to and significantly higher than those of laboratory strains of *V. parahaemolyticus* PSU.SCB.16S.14. As shown in Figure 5A, all the 50 isolates were swimmers; 7 isolates were weak swimmers (< 15 mm); 26 isolates were moderate swimmers (< 30 mm); and 17 isolates were strong swimmers (> 30 mm). Similarly, all isolates were swarmers (Figure 5B). Among all isolates, 43 isolates were moderates swarm cells; and 7 isolates were strong swarm cells. Thus, all isolates showed a relatively strong mobility. *V. parahaemolyticus* could move via propelling with the aid of flagella. Swimming and swarming behaviors are initial requirement for biofilm formation [49]. All *V. parahaemolyticus* isolates had relatively strong mobility, associated with their biofilm formation.

#### 3.2.4. Biofilm Formation Capacity

The bacterial biofilm protects pathogens from environmental stress such as antimicrobial and increases disease severity in infected host [50,51,52]. The biofilm was formed by 50 isolates when tested using the CV staining (Figure 5C). *V. parahaemolyticus* was able to form biofilms and attached to the surfaces of seafood [53]. Sun et al. [54] found that *V. parahaemolyticus* isolated from stool specimens of diarrhea patients exhibited biofilm formation. All clinical *V. parahaemolyticus* isolates were biofilm producers. Biofilm formation is governed by the source of isolates and cultural temperature. In general, pathogenic isolates produced more biofilms than non-pathogenic counterpart [55,56]. Optimum temperature for biofilm formation by *V. parahaemolyticus* was 37 °C [57]. In general, bacterial cells entrapped in biofilms are more resistant to harsh conditions [53].

### 3.3. Antimicrobial Activity of COS-Tea Polyphenol Conjugates toward V. parahaemolyticus Isolates

Antimicrobial effects of COS-tea polyphenol conjugates on clinical and Asian green mussel *V. parahaemolyticus* isolates were examined. Antimicrobial activity was expressed as MIC and MBC of COS-tea polyphenol conjugates against 50 *V. parahaemolyticus* isolates. COS-tea polyphenol conjugates showed the adverse effect on the growth of *V. parahaemolyticus* isolated from clinical and Asian green mussel samples. COS–CAT had MIC (0.128–1.024 mg/mL) and MBC (0.256–2.048 mg/mL), whereas COS–EGCG possessed MIC (0.032–0.128 mg/mL) and MBC (0.256–2.048 mg/mL). The MBC/MIC ratio has been used to determine the antibacterial potential of substances. MBC lower than 1.024 mg/mL was observed for both COS-CAT and COS-EGCG. COS-CAT showed a MBC/MIC ratio of ≤ 4 against 19 tested isolates including M1, M3, M21, M26, M42, M45, M47, M71, M72, M77, M89, M91, M92, M95, M106, M112, M121, HVP1, and HVP 2. COS-EGCG had a MBC/MIC ratio of ≤ 4 toward seven tested isolates involving M3, M21, M26, M89, M109, M121, and HVP2. The COS-CAT showed a MBC/MIC ratio of ≤ 8 against eight tested isolates, which included M6, M37, M43, M52, M58, M95, HVP3, and HVP5, whereas COS- EGCG had a MBC/MIC ratio of ≤ 8 against 14 tested isolates, which consisted of M1, M42, M77, M78, M79, M90, M91, M92, M95, M106, M112, M120, HVP1 and HVP5. COS-polyphenol conjugates have the potential to control foodborne pathogens [20,21]. Recently, Mittal et al. [21] reported that COS-CAT conjugate showed higher antioxidant and antimicrobial activities than COS and other COS-polyphenol conjugates. COS-CAT showed antimicrobial activity against both Gram-negative and Gram-positive bacteria. Antimicrobial activity of COS and polyphenols was linked to bacterial cell wall disintegration. Furthermore, changes in microbial DNA, mRNA, and protein synthesis via diffused COS and polyphenols could bring about the cell death [20,21]. Blueberry extract showed stronger antimicrobial effect on *V. parahaemolyticus*, which had no virulence genes than *V. parahaemolyticus* ATCC17802 (*tdh*^−^/*trh*^+^) and ATCC 33847 (*tdh*^+^/*trh*^−^), which had virulence genes. Increased virulence was associated with augmented antibiotic resistance [41]. COS-tea polyphenol conjugates as a bactericidal/bacteriostatic substance might have stronger antibacterial activity against a virulent strain. Antimicrobial activity of COS polyphenol conjugates against *V. parahaemolyticus* PSU.SCB.16S, MIC and MBC were 32 μg/mL and 64 μg/mL for the COS-CAT, respectively; and MIC and MBC of 64 μg/mL and 128 μg/mL were recorded for the COS-EGCG, respectively [21]. Gram-negative bacteria generally have hydrophilic thin outer membrane, comprising lipopolysaccharides. Therefore, they are susceptible to cellular lysis via COS and its polyphenol conjugates [58]. COS-tea polyphenol, especially COS-CAT, was a promising antimicrobial agent toward both spoilage and pathogenic bacteria. Sun et al. [59] reported that *tolC* gene expression was downregulated in *V. parahaemolyticus* F13. Although MIC and MBC of COS-CAT were higher than those of some antibiotics, it had the efficacy in inhibiting both pathogenic and non-pathogenic *V. parahaemolyticus*. Overall, non-pathogenic *V. parahaemolyticus* generally had more sensitivity to antibiotics than pathogenic *V. parahaemolyticus*. However, pathogenic and non-pathogenic *V. parahaemolyticus* were similarly susceptible to COS-CAT in the present study.

### 3.4. Antibiotic Susceptibility Profile of Different V. parahaemolyticus Isolates

Eight *V. parahaemolyticus* isolates represented *V. parahaemolyticus* PSU.SCB.16S.14 (Laboratory strain, VP), Asian green mussel from farm (M1 isolate), Asian green mussel from natural habitat (M42 isolate), Asian green mussel from local markets (M77, M91, M92, and M106 isolates), and clinical sample of stool specimens of diarrhea patients (HVP1 isolate) were used for testing. MBCs of COS-tea polyphenol conjugates against all *V. parahaemolyticus* (8 strains) were 1.024 mg/mL as shown in Table 3. Varying antibiotic susceptibility profiles with 21 antibiotics toward those eight isolates were noticeable (Table 4). Seven antibiotics namely ampicillin, cefoxitin, ceftriaxone, colistin, doripenem, ertapenem, and netilmicin did not show susceptible, intermediate, and resistant results because CLSI breakpoints of these antibiotics did not exist for *V. parahaemolyticus* [60]. Isolates tested were highly susceptible to antibiotics such as Amikacin (100%), ciprofloxacin (100%), gentamicin (100%), imipenem (100%), and levofloxacin (100%). However, HPV1 clinical isolate showed high resistance to amoxicillin/clavulanic acid, ampicillin/sulbactam, cefepime, cefotaxime, ceftazidime, cefuroxime, meropenem, piperacillin/tazobactam, and trimethoprim/sulfamethoxazole with MAR index of 0.64. This isolate was resistance to 9 antibiotics of 14 antibiotics tested. Most of the six Asian green mussel *V. parahaemolyticus* isolates in this study exhibited high sensitivity to all antibiotics, but M42 isolate exhibited intermediate resistance to cefuroxime. Elexson et al. [57] found that all *V. parahaemolyticus* isolates from cultured seafood products were resistant to penicillin and ampicillin. However, it has been discovered that the Asian green mussel cultivated in Thailand is frequently an open system culture in coasts and estuaries, in which antibiotics are not required. As a result, no drug resistant *V. parahaemolyticus* isolated from natural Asian green mussel farms and Asian green mussels sold in the local market was found in this study. Another health risk may arise with cross-contamination by shellfish to other seafoods in the market. To address the potential consequences of pathogenic *V. parahaemolyticus* in seafood, continuous monitoring of environmental and seafood samples, including mussel as well as tracking the source of clinical and environmental strains are still needed.

## 4. Conclusions

Fifty collected isolates including Asian green mussel samples, clinical and laboratory strains were identified as *V. parahaemolyticus* based on their morphological, biochemical, and molecular characteristics. They were all biofilm producers with strong motile ability. Only six isolates (12%) from the clinical sample were positive for the virulence *tdh* gene (*tdh*^+^*trh*^−^) and had a positive result for the KP test. COS-CAT demonstrated the greatest bactericidal action against *V. parahaemolyticus* isolated from Asian green mussels and clinical samples with an MBC value of 1.024 mg/mL. In addition, *V. parahaemolyticus* isolated from Asian green mussel farms, natural habitat, and local markets showed no antibiotic resistance. Only the sample clinical isolates had a MAR value of 0.64 and were extremely resistant to nine kinds of antibiotics. Hence, to address the potential consequences of pathogenic *V. parahaemolyticus* in seafood, constant monitoring of environmental and seafood samples is still essential for food safety assurance.

## Figures and Tables

**Figure 1 foods-11-04048-f001:**
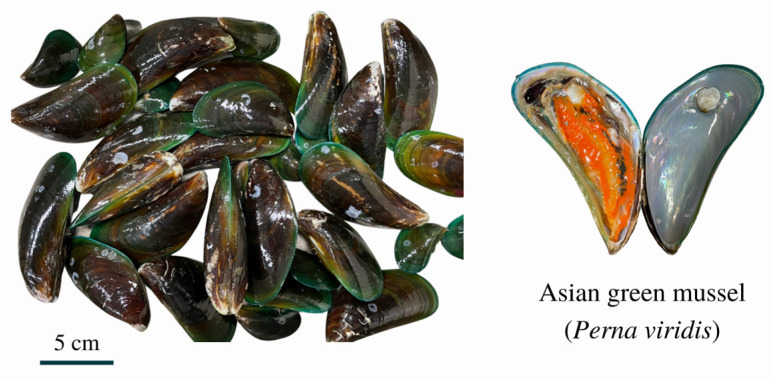
Asian green mussel (*Perna viridis*) collected from the south of Thailand.

**Figure 2 foods-11-04048-f002:**
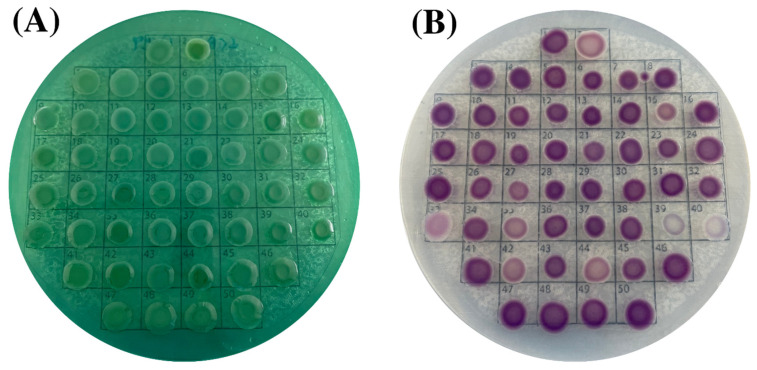
Colony morphology of fifty *V. parahaemolyticus* isolates on thiosulphate citrate bile salts sucrose agar plate (**A**) and CHROMagar^TM^ Vibrio agar plate (**B**).

**Figure 3 foods-11-04048-f003:**
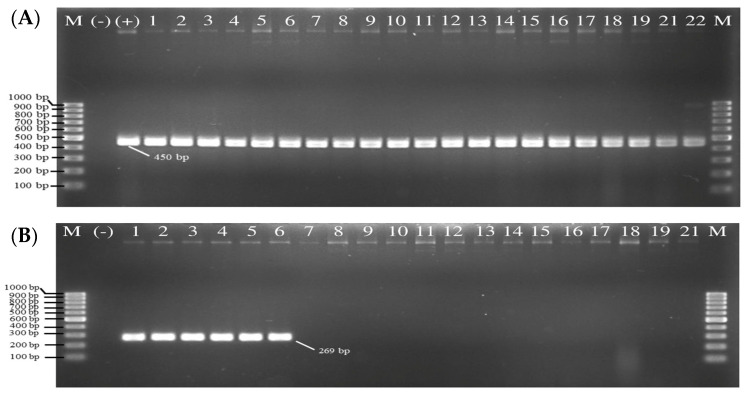
Gel electrophoresis of products of *tlh* primer (**A**) (Lane M: DNA marker, Lanes 1–22: representative samples, Lane (−): negative-control (DNA free template), Lane (+): positive-control (*V. parahaemolyticus* PSU.SCB.16S.14) and products of *tdh* primer (**B**) Lane M: DNA marker, Lanes 1–6: representative isolates of pathogenic *V. parahaemolyticus*, Lanes 7–21: representative isolates of non-pathogenic *V. parahaemolyticus*, Lane (−): negative-control (DNA free template), Lane (+).

**Figure 4 foods-11-04048-f004:**
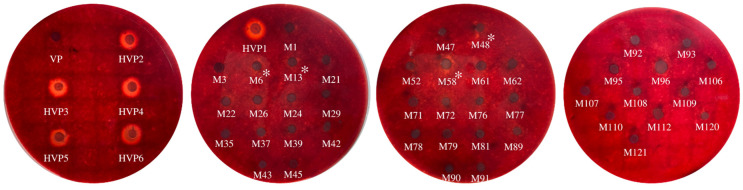
The hemolytic activity of *V. parahaemolyticus* isolates against RBCs. β-hemolysis zone surrounding the spot of growth on the Wagatsuma agar plate was measured. *V. parahaemolyticus* cells were grown on Wagatsuma agar for 24 h. * Isolates with weak hemolysis.

**Figure 5 foods-11-04048-f005:**
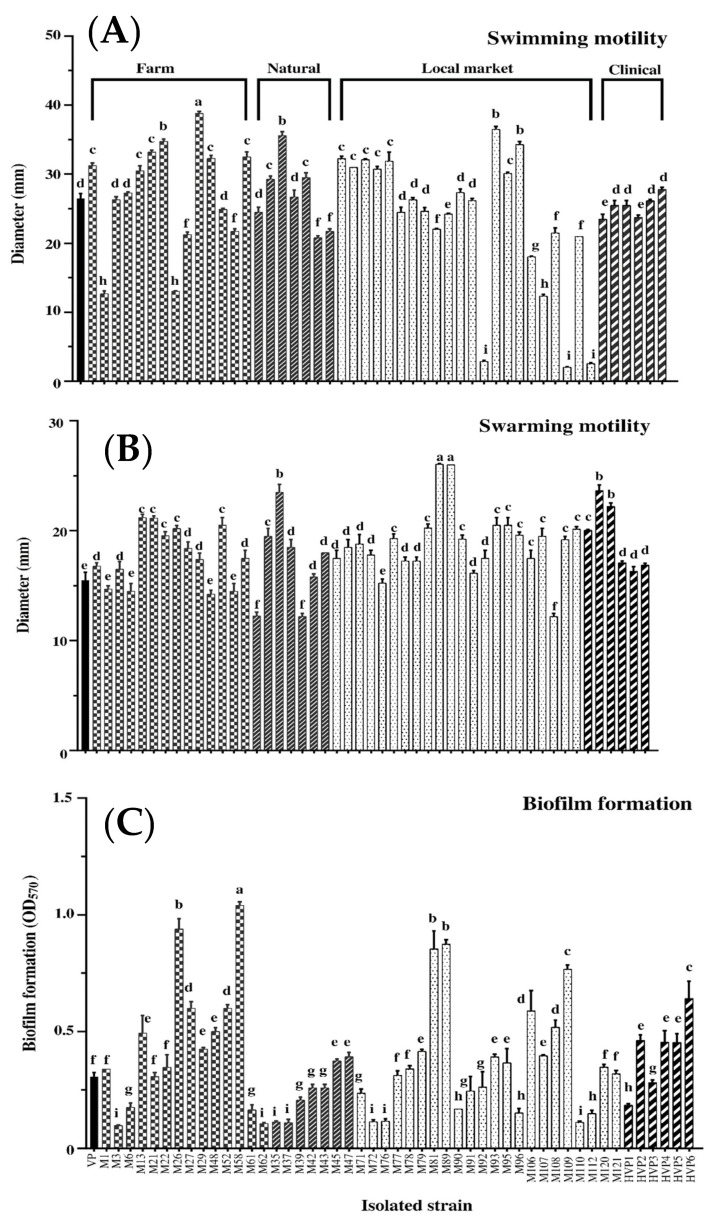
Swimming motility (**A**), swarming motility (**B**), and biofilm formation ability (**C**) of 50 isolates of *V. parahaemolyticus*. Different lowercase letters on the bars denote the significant differences (*p* < 0.05).

**Table 1 foods-11-04048-t001:** Provinces, types of places, and number of *V. parahaemolyticus* isolates collected from the south of Thailand.

Provinces	Types of Places Collected	Number of Samples
Suratthani	Local market	2
	Asian green mussel farm	6
	Natural habitat	4
Trang	Local market	2
	Asian green mussel farm	4
	Natural habitat	2
Songkhla	Local market	8

**Table 2 foods-11-04048-t002:** Primers selected for the detection of virulence genes of *V. parahaemolyticus*.

Primer		Sequence (5′-3′)	Amplicon Size (bp)	Reference
*tlh*	F	AAA GCG GAT TAT GCA GAA GCA CTG	450	Siddique et al. [27]
	R	GCT ACT TTC TAG CAT TTT CTC TGC		
*tdh*	F	CCA TCT GTC CCT TTT CCT GCC	269	Siddique et al. [27]
	R	CCA CTA CCA CTC TCA TAT GC		
*trh*	F	TTG GCT TCG ATA TTT TCA GTA TCT	500	Siddique et al. [27]
	R	CAT AAC AAA CAT ATG CCC ATT TCC G		

**Table 3 foods-11-04048-t003:** Minimum inhibitory concentration (MIC) and minimum bactericidal concentration (MBC) of COS-tea polyphenol conjugates against eight isolates of *V. parahaemolyticus*.

Bacterial Strains	COS-CAT (mg/mL)		COS-EGCG (mg/mL)	
	MIC	MBC	MIC	MBC
VP PSU.SCB.16S.14	0.256	1.024	0.064	1.024
M1	0.256	1.024	0.128	1.024
M42	0.256	1.024	0.128	1.024
M77	0.256	1.024	0.128	1.024
M91	0.256	1.024	0.128	1.024
M92	0.256	1.024	0.128	1.024
M106	0.256	1.024	0.128	1.024
HVP1	0.256	1.024	0.128	1.024

**Table 4 foods-11-04048-t004:** Twenty-one antibiotics resistance profiles of *V. parahaemolyticus* isolates from Asian green mussel, clinical, and laboratory samples.

Antibiotics	Concentration (μg/mL)	VP	M1	M42	M77	M91	M92	M106	HVP1
Amikacin	8–32	S	S	S	S	S	S	S	S
Amoxicillin/Clavulanic acid	4/2–16/8	S	S	S	S	S	S	S	R
Ampicillin	8–16	NI	NI	NI	NI	NI	NI	NI	NI
Ampicillin/Sulbactam	4/2–16/8	S	S	S	S	S	S	S	R
Cefepime	1–32	S	S	S	S	S	S	S	R
Cefotaxime	1–32	S	S	S	S	S	S	S	R
Cefoxitin	4–16	NI	NI	NI	NI	NI	NI	NI	NI
Ceftazidime	1–32	S	S	S	S	S	S	S	R
Ceftriaxone	0.5–32	NI	NI	NI	NI	NI	NI	NI	NI
Cefuroxime	8–16	S	S	I	S	S	S	S	R
Ciprofloxacin	0.06–2	S	S	S	S	S	S	S	S
Colistin	1–8	NI	NI	NI	NI	NI	NI	NI	NI
Doripenem	0.5–16	NI	NI	NI	NI	NI	NI	NI	NI
Ertapenem	0.5–4	NI	NI	NI	NI	NI	NI	NI	NI
Gentamicin	2–8	S	S	S	S	S	S	S	S
Imipenem	0.5–16	S	S	S	S	S	S	S	S
Levofloxacin	0.06–8	S	S	S	S	S	S	S	S
Meropenem	0.5–16	S	S	S	S	S	S	S	R
Netilmicin	8–16	NI	NI	NI	NI	NI	NI	NI	NI
Piperacillin/Tazobactam	8/4–64/4	S	S	S	S	S	S	S	R
Trimethoprim/Sulfamethoxazole	1/19–4/76	S	S	S	S	S	S	S	R

S: Susceptible; I: Intermediate; R: Resistant; NI: No interpretation. Note: Eight *V. parahaemolyticus* isolates included *V. parahaemolyticus* PSU.SCB.16S.14 (VP), isolates from Asian green mussel from farm (M1 isolate), Asian green mussel from natural habitat (M42 isolate), Asian green mussel from local markets (M77, M91, M92, and M106 isolates) and clinical isolate from stool specimens of diarrhea patients (HVP1 isolate).

## Data Availability

Data is contained within the article or supplementary material.

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
