# Peer review of "Vibrio parahaemolyticus Isolates from Asian Green Mussel: Molecular Characteristics, Virulence and Their Inhibition by Chitooligosaccharide-Tea Polyphenol Conjugates"

_foods, 2022, doi:10.3390/foods11244048_

Round 1

Reviewer 1 Report

The manuscript by Palamae et al. describes the characteristics of several Vibrio isolates (from asian green mussels), presumably identified as V. parahaemolyticus, as well as the antagonistic effect of COS-CAT.

The identification was based on colony phenotype and MALDI analysis; however, molecular characterization (DNA sequencing of relevant genes validated for Vibrio identification, followed by proper bioinformatic analyses, e.g., concatenation analysis) is missing, which strongly weakens the entire manuscript. Overall, the paper is dominated by characterization at the level of presumptive virulence and biochemical features, along with antimicrobial sensitivity tests.

Minor comments: the English language should be extensively revised, and the Latin names should be italicized throughout the text.

Author Response

Author's Reply to the Review Report

REVIEWER 1

Comments and Suggestions for Authors

The manuscript by Palamae et al. describes the characteristics of several Vibrio isolates (from asian green mussels), presumably identified as V. parahaemolyticus, as well as the antagonistic effect of COS-CAT.

The identification was based on colony phenotype and MALDI analysis; however, molecular characterization (DNA sequencing of relevant genes validated for Vibrio identification, followed by proper bioinformatic analyses, e.g., concatenation analysis) is missing, which strongly weakens the entire manuscript. Overall, the paper is dominated by characterization at the level of presumptive virulence and biochemical features, along with antimicrobial sensitivity tests.

*****Authors do agree with the reviewer regarding the required advanced techniques like DNA sequencing as well as bioinformatic analysis, etc. for further identification of the selected strains. However, the present study carried out the identification based on colony phenotype and MALDI analysis, which have been commonly used for identification. In addition, we used gene tld for confirmation of specific species for V. parahaemolyticus, following the method of ISO. Apart from aforementioned identification, another goal of our study was to chitooligosaccharide-catechin conjugate as the natural antimicrobial agent to inhibit V. parahaemolyticus. The mode of action was also elucidated. Therefore, all the comments regarding the molecular identification is taken into consideration for our next study. Author would like to thank the reviewer for the insightful comment and invaluable suggestion.

COMMENTS 1: “The English language should be extensively revised, and the Latin names should be italicized throughout the text.”

RESPONSE 1: Thank you for the suggestion in English improvement. Authors have checked English using “Grammarly’ software throughout the manuscript. For Latin names, we have checked and edited, in which scientific name and gene name have been written using italic. Please see lines 16-31, 36-87, 175-178, 217-245, 253-255, 268, 359, 372, 405 and 415-426. 

Reviewer 2 Report

I highly appreciate the concept, methodology and results (with their discussion) of this original research, with real practical importance, written by authors well experienced in the field. Chitooligosaccharide-tea polyphenol, especially Chitooligosaccharide-catechin conjugate was proven to have a high antibacterial activity against V. parahaemolyticus isolated from Asian green mussels and clinical samples (stool from patients with diarrhea). Even though the isolates from mussel showed susceptibility to most antibiotics, this study has huge relevance, as antibiotic-resistant bacteria have been a major issue across the world and this resistance is increasing. It was interesting to see the similarities and differences of the isolates from Asian Green Mussel vs those from patients with diarrhea and from laboratory. Minor comments/questions:

1. Abstract:

a. 1st sentence – Please define that was your aim and I would include here also the antimicrobial activity of COS polyphenol conjugates (in fact, this is the most important aspect).

b. Please also clarify: In the whole study it is mentioned, as in the first sentence that there were fifty isolates (and I analyzed carefully the figures as well). But, in the next sentence of the Abstract, it is written - line 16:Fifty-three isolates were randomly selected from 520 isolates” – contrary to what appears in Material and Methods - line 103, where it is written “Forty-three isolates with different colony colors from the total 520 isolates”.  So, please clarify: 43 isolates from mussel, 6 isolates from stool of patients with diarrhea and 1 from the lab. Correct?

c. Please also define the abbreviation COS (chitooligosaccharide), before its 1st use, as well as MAR (multiple antibiotic resistance).

2. Introduction: Please revise the following sentence, as some words are missing: “Although COS polyphenol conjugates have been reported to be effective against a variety of microorganisms.” – lines 80-81. Also, the next one: “Nevertheless, no reports on its antimicrobial activity against V. parahaemolyticus isolated from Asian green mussel and clinical sample” – lines 81-83. Please also clarify the next sentence and correct the language: “In additional, drug sensitivity, virulence and molecular characteristics of V. parahaemolyticus isolated from Asian green mussel collected from the south of Thailand were also elucidated.” Basically, all these three sentences could be rewritten, in order to formulate a proper aim of this study. Or, in any case, please describe clearly the aim of your research, not what was elucidated.

3. Results – should be Results and Discussion:

a. Line 215: there are 59 isolates mentioned now – “All fifty-nine collected isolates from Asian green mussel samples, clinical and laboratory strains were identified as V. parahaemolyticus”. Please clarify.

b. Please insert strength and limitations of your research.

 4. Figure 1. Please remove “This is a figure. Schemes follow the same formatting.” from the title

Author Response

Author's Reply to the Review Report

REVIEWER 2

Comments and Suggestions for Authors

“I highly appreciate the concept, methodology and results (with their discussion) of this original research, with real practical importance, written by authors well experienced in the field. Chitooligosaccharide-tea polyphenol, especially Chitooligosaccharide-catechin conjugate was proven to have a high antibacterial activity against V. parahaemolyticus isolated from Asian green mussels and clinical samples (stool from patients with diarrhea). Even though the isolates from mussel showed susceptibility to most antibiotics, this study has huge relevance, as antibiotic-resistant bacteria have been a major issue across the world and this resistance is increasing. It was interesting to see the similarities and differences of the isolates from Asian Green Mussel vs those from patients with diarrhea and from laboratory. Minor comments/questions:”

******Authors would like to express the sincere thank to the reviewer for understanding in our work. All the queries have been responded and the correction have been done via track changes and line numbers have also been given.

COMMENT 1: “Abstract:

  1. 1stsentence – Please define that was your aim and I would include here also the antimicrobial activity of COS polyphenol conjugates (in fact, this is the most important aspect).

*****Authors do agree with the reviewer for including antimicrobial activity of COS-polyphenol conjugates, which was another major goal of this study in the abstract. Please see the second sentence. See line 17-18.

  1. Please also clarify: In the whole study it is mentioned, as in the first sentence that there were fifty isolates (and I analyzed carefully the figures as well). But, in the next sentence of the Abstract, it is written - line 16:“Fifty-three isolates were randomly selected from 520 isolates” – contrary to what appears in Material and Methods - line 103, where it is written “Forty-three isolates with different colony colors from the total 520 isolates”.  So, please clarify: 43 isolates from mussel, 6 isolates from stool of patients with diarrhea and 1 from the lab. Correct?

*****Sorry for the mistake. The comment from the reviewer is correct. The number of isolates has been edited to 43 isolates from mussel samples. Please see line 18.

  1. Please also define the abbreviation COS (chitooligosaccharide), before its 1stuse, as well as MAR (multiple antibiotic resistance).”

****The COS and MAR have been defined at the first time. See lines 17 and 29-30, respectively.

COMMENT 2: “Introduction: Please revise the following sentence, as some words are missing: “Although COS polyphenol conjugates have been reported to be effective against a variety of microorganisms.” – lines 80-81. Also, the next one: “Nevertheless, no reports on its antimicrobial activity against V. parahaemolyticus isolated from Asian green mussel and clinical sample” – lines 81-83. Please also clarify the next sentence and correct the language: “In additional, drug sensitivity, virulence and molecular characteristics of V. parahaemolyticus isolated from Asian green mussel collected from the south of Thailand were also elucidated.” Basically, all these three sentences could be rewritten, in order to formulate a proper aim of this study. Or, in any case, please describe clearly the aim of your research, not what was elucidated.”

*****Those sentences have been rewritten for better clarity. Please see lines 82-87.

COMMENT 3: “Results – should be Results and Discussion:

****Correction has been done. Please see line 217.

  1. Line 215: there are 59 isolates mentioned now – “All fifty-nine collected isolates from Asian green mussel samples, clinical and laboratory strains were identified as V. parahaemolyticus”. Please clarify.

****Sorry for our mistake. In fact, the true number of isolates was 50. The correction has been made properly. Please see line 219.

  1. Please insert strength and limitations of your research.”

*****Thank you so much for the comment. This is the first report on the occurrence of V. parahaemolyticus in Asian green mussel cultured in Thailand. This is the strength of this study, providing the basic information on V. parahaemolyticus in mussel and the inhibition by COS-polyphenol conjugate. Those have been mentioned in Introduction (lines 83-87).

However, the limitation of this research was that the experiments were carried out in a few areas (provinces) of southern Thailand. Therefore, the collecting sites must be extended to cover most of southern part of Thailand to acquire the reliable and useful data.  Authors have included the limitation in lines 244-245.

The further suggestion had already been provided in lines 401-403 as follows:  To address the potential consequences of pathogenic V. parahaemolyticus in seafood, continuous monitoring of environmental and seafood samples, including mussel as well as tracking the source of clinical and environmental strains are still needed.

COMMENT 4: “Figure 1. Please remove “This is a figure. Schemes follow the same formatting.” from the title”

*****Sorry. Authors have cross-checked and deleted the irrelevant phrase. Thank you so much. Please see line 117.

Round 2

Reviewer 1 Report

The revised manuscript is nearly unchanged compared to the first version, except for some minor corrections.

Though the PCR positivity to tdh amplification can be considered a confirmation of the species, I wonder why the authors claim they will include molecular analyses in future papers.

Last, the manuscript does not report the elucidation of mechanisms involved in antimicrobial activity; whereas, the mechanisms hypothesized and previously described in the literature are reported.